# Exploring the Impact of COVID-19 Response on Population Health in Saudi Arabia: Results from the “Sharik” Health Indicators Surveillance System during 2020

**DOI:** 10.3390/ijerph18105291

**Published:** 2021-05-16

**Authors:** Nasser F. BinDhim, Nora A. Althumiri, Mada H. Basyouni, Norah AlMousa, Mohammed F. AlJuwaysim, Alanoud Alhakbani, Najat Alrashed, Elaf Almahmoud, Rawan AlAloula, Saleh A. Alqahtani

**Affiliations:** 1Sharik Association for Health Research, Riyadh 13326, Saudi Arabia; nasser.bindhim@sharikhealth.net (N.F.B.); mada.basyouni@sharikhealth.net (M.H.B.); 2170001797@iau.edu.sa (N.A.); 216006361@student.kfu.edu.sa (M.F.A.); 2College of Medicine, Alfaisal University, Riyadh 11533, Saudi Arabia; 3Saudi Food and Drug Authority, Riyadh 13513, Saudi Arabia; 4Ministry of Health, Riyadh 11176, Saudi Arabia; 5College of Public Health, Imam Abdulrahman Bin Faisal University, Dammam 31441, Saudi Arabia; 6College of Pharmacy, King Faisal University, AlAhsa 31982, Saudi Arabia; 7Center for Complex Engineering Systems, King Abdulaziz City for Science and Technology, Riyadh 12354, Saudi Arabia; alhakbani@kacst.edu.sa (A.A.); nalrashed@kacst.edu.sa (N.A.); e.almahmoud@cces-kacst-mit.org (E.A.); raloula@kacst.edu.sa (R.A.); 8King Faisal Specialist Hospital and Research Centre, Liver Transplant Unit, Riyadh 11211, Saudi Arabia; salqaht1@jhmi.edu; 9Division of Gastroenterology and Hepatology, Johns Hopkins University, Baltimore, MD 21218, USA

**Keywords:** population health, Saudi Arabia, COVID-19, surveillance, risk factors, chronic diseases

## Abstract

Background: Although some studies have explored the effects of responses to COVID-19 on mortality, there are limited data on their effects on more immediate health risk factors and the trends of chronic diseases. Objective: To explore the prevalence of some behavioral health risk factors, intermediate risk factors, and chronic diseases at different timepoints during 2020 using the data available from a currently used surveillance system in Saudi Arabia. Methods: This study undertook a secondary analysis of data from the Sharik Health Indicators Surveillance System (SHISS). The SHISS employs short cross-sectional phone interviews, conducted in all 13 administrative regions of Saudi Arabia on a quarterly basis. Each interview lasts approximately 4 min and is conducted by a trained data collector. The SHISS collects demographic data, as well as data on the major behavioral and intermediate chronic disease risk factors and the major chronic diseases, including diabetes, heart disease, stroke, cancer, and chronic respiratory diseases. Results: Of the 44,782 potential participants contacted in 2020, 30,134 completed the interview, with a response rate of 67.29%. Out of the total participants, 51.2% were female. The mean age was 36.5. The behavioral risk factors in this period exhibited significant changes compared to those in the first quarter (Q1) of 2020, when there were no significant restrictions on daily activities. These significant changes are related to reductions in fruit and vegetable intake (adjusted odds ratio (AOR), 0.23) and physical activity (AOR, 0.483), and a significant increase in e-cigarette use (AOR 1.531). In terms of the intermediate risk factors observed in the SHISS, significant increases in hypercholesterolemia (AOR, 1.225) and hypertension (AOR, 1.190) were observed. Finally, heart disease (AOR, 1.279) and diabetes (AOR, 1.138) displayed significant increases compared to Q1. Conclusions: This study shows some evidence of the impact of responses to COVID-19 on the health of the population in Saudi Arabia. Significant reductions in fruit and vegetable intake and physical activity, and significant increases in e-cigarette use, hypertension, and hypercholesterolemia may increase the burden of chronic diseases in Saudi Arabia in the near future. Thus, continuous monitoring of the health risk factors within the population, and early interventions, are recommended to prevent future increases in chronic diseases.

## 1. Introduction

Almost all countries have implemented restrictions on movement in response to the COVID-19 pandemic, with the aim of cutting the transmission of the virus by reducing close contact between individuals; however, these measures have had profound consequences on social, economic, and health behaviors [1]. The social distancing measures have included the following: advising the whole population to self-isolate if they or members of their household have symptoms; bans on social gatherings (including mass gatherings) and halting flights, public transport and movement between cities; closure of “non-essential” workplaces (anything outside the health and social care sector, utilities, and the food chain) with encouraged working from home for those who can; the closure of sports and fitness centers, schools, colleges, and universities; the prohibition of all “non-essential” movement [1]. These measures were rolled out in order to slow the spread of the virus and reduce the risk of health services becoming overwhelmed [1]. The COVID-19 pandemic has changed many social, economic, environmental, and healthcare systems, and impacted determinants of health [2].

There are several mechanisms through which responses to the COVID-19 pandemic are likely to affect health, including economic effects, social isolation, family relationships, health-related behaviors, disruption to essential health services, and psychosocial stress [1]. For instance, excess mortality could result from healthcare avoidance, diagnosis and treatment delays, or insufficient care for other urgent conditions, owing to a reduced capacity to treat other medical emergencies [1,3]. On the other hand, other causes of mortality may be temporarily reduced, such as deaths from acute respiratory conditions or traffic accidents, but these mortality reductions may be outweighed by the excesses [3]. One study estimated a 0.9-year decline in annual life expectancy in Spain, a reduction in life expectancy not seen for decades [3]. England, Wales and Spain experienced the largest effects: ~100 excess deaths per 100,000 people, equivalent to a 37% (30–44%) relative increase in England and Wales and a 38% (31–45%) relative increase in Spain [3]. Bulgaria, New Zealand, Slovakia, Australia, Czechia, Hungary, Poland, Norway, Denmark, and Finland also experienced potential mortality alterations, ranging from small reductions to increases of 5% or less in either sex [3].

The indirect health effects of the COVID-19 pandemic and the responses to it can also be regarded as substantial [1,2]. The indirect effects may include denied or delayed disease prevention and medical procedures for acute and chronic conditions; reductions in awareness and educational public health campaigns for noncommunicable diseases; reduced physical activity [4,5]; losses of jobs and income and reductions in overall living status; disruption of social networks; increases in self-harm and anxieties over contracting the disease [4]; changes in the quantity and quality of food and overall diet; use of tobacco, alcohol and other drugs; changes in other infectious disease rates [1,2]. Such changes in the behavioral and intermediate risk factors related to noncommunicable diseases could significantly increase or decrease their prevalence in the population.

In Saudi Arabia, on 2 March 2020, the government reported the first case of COVID-19 [4]. As COVID-19 continued to spread, the Saudi government enforced many drastic measures to curb the spread of the disease, including partial (usually between 3 p.m. and 6 a.m.) and 24 h lockdowns, suspensions of religious activities such as prayer in mosques and Umrah mass gatherings, and a complete lockdown during the Eid holidays [4]. Consequently, as in many countries, the economic impact of the lockdown affected many businesses in Saudi Arabia, leading to job losses or reductions in income [4]. The whole period of complete and partial lockdown lasted around three months, between mid-March 2020 and the end of May 2020, and for some cities and businesses the lockdown continued until the end of June 2020 [4]. Working from home was encouraged for government sector employees until August 2020, and at the time of writing is still being encouraged for some subgroups or organizations with small workspaces, in order to facilitate social distancing. As such, the strictest responses to COVID-19 occurred in the second quarter of 2020, while the responses in the third quarter were generally less strict.

Public health surveillance is one of the keystones of public health practice, empowering decision-makers to manage public health more effectively by providing timely and useful data and evidence [5]. The Sharik Health Indicators Surveillance System (SHISS) is a multi-wave, nationwide, survey-based population health surveillance program that was initiated in early 2020 by the Sharik Association for Health Research [6]. The SHISS was developed to capture timely population health data, so as to provide decision-makers with important health indicators more frequently, and to help monitor the population’s health over time. Such data can help in understanding the changes in population health indicators and the effects of major events or national public health programs on these indicators. The descriptive results of the SHISS are disseminated onto an electronic statistical dashboard within a week of each wave’s completion, and then communicated to the stakeholders. Such data can be used to explore the effects of major events such as COVID-19.

Although some studies have explored the effects of responses to COVID-19 on mortality, we were not able to identify any studies exploring the effects of these on more immediate risk factors or the trends of chronic diseases. Thus, this study explores the prevalence of certain behavioral health risk factors, intermediate risk factors, and chronic diseases at different timepoints during 2020, using the data available from SHISS.

## 2. Method

### 2.1. Design

This study undertook a secondary analysis of data from the SHISS. The SHISS consists of short cross-sectional phone interviews conducted in all 13 administrative regions of Saudi Arabia on a quarterly basis. Each interview lasts approximately 4 min and is conducted by a trained data collector. In this study, we used the QPlatform^®^ data collection system [7], which has integrated eligibility and sampling modules, to control the distribution of the sample and to prevent human-related sampling bias [6]. All questions had to be answered for an individual questionnaire to be successfully submitted to the database. All data were coded and stored on the QPlatform^®^ database [7].

### 2.2. Sampling and Sample Size

The SHISS used the proportional quota sampling technique to obtain an equal distribution of participants, stratified by age and gender, within and across the 13 administrative regions of Saudi Arabia. We used two age groups, based on the Saudi Arabian median age of 36 years. This established a quota of 52 for this study.

The sample size was calculated based on a medium effect size of nearly 0.3, with 80% power and a 95% confidence level to allow for comparison between quota on region, age, and gender level [8]. Thus, each quota required at least 100 participants, and a total sample of 400 per region, for a total of 5200 participants/wave. Once the quota sample was reached, we stopped accepting participants with similar characteristics into the study. Quota sampling is an automated process controlled by the data collection system, with no human interference [7].

In 2020, extra waves of questionnaires were conducted in the second quarter to generate a monthly sample between mid-March and June 2020, so as to capture more data for the peak of the COVID-19 pandemic, which meant a total of seven waves in 2020. The last wave of sampling in the fourth quarter was undertaken in December 2020.

### 2.3. Participants and Recruitment

Participant recruitment was limited to Arabic-speaking Saudi residents who were ≥18 years old. A random mobile phone number list was generated by the Sharik Association for Health Research to identify potential participants [9]. The Sharik database is composed of individuals who are interested in participating in future research projects, and contains more than 80,000 users distributed across the 13 regions of Saudi Arabia and continues to grow [9]. Participants were contacted by phone on up to three occasions. If they did not respond, a new potential participant with similar demographics was extracted from the database; this process was repeated until the quota was completed and closed automatically. After obtaining consent to participate, the interviewer assessed the eligibility, and if the participant was eligible, the reviewer started the interview.

### 2.4. Questionnaire Design and Validation

After providing verbal consent, participants were asked their age and region to determine eligibility. Then, the data collector recorded the age, gender, and region of the participant. Then, any major chronic disease they had, and their main behavioral and intermediate risk factors (as suggested by the World Health Organization (WHO) and the CDC) were assessed [10,11]. As shown in the SHISS data model (Figure 1), the dataset includes behavioral risk factors (diet, physical activity and tobacco use, including cigarettes, waterpipes, and e-cigarettes), diagnosed on-treatment intermediate risk factors (hypertension and hypercholesterolemia), obesity measured as a body mass index (BMI) via height and weight, and finally diagnosed major chronic diseases that the participant was currently receiving treatment for, including diabetes, heart disease, stroke, cancer, and chronic respiratory disease. Finally, the presence of a diagnosed genetic disease was also recorded as a nonmodifiable risk factor.

The participants were asked to provide their height in cm and weight in kg. We also calculated the participants’ body mass indices (BMIs). We used the Center for Disease Control and Prevention’s (CDC) BMI category status, which specifies a BMI of 30 or above as obese [12]. This study used the WHO’s global recommendations of physical activity for adults (18–64 years old): (1) vigorous intensity physical activity (VIPA) for 75 min per week, or (2) moderate intensity physical activity (MIPA) for 150 min per week [13]. Based on the participants’ self-reported responses to the interview questionnaire (i.e., number of exercise minutes, frequency, and intensity level per week), two categorical outcome variables were created that reflected whether or not guidelines were met: an acceptable level of physical activity (ALPA) (at least 150 min of MIPA per week and/or at least 75 min of VIPA per week) and a low level of physical activity (LLPA) (less than 150 min of MIPA and/or less than 75 min of VIPA).

We asked the participants about their daily fruit and vegetable intake. If a participant’s daily food intake included at least one portion of fruit and one portion of vegetables, they were categorized as having an acceptable level of fruit and vegetable intake (AFVI). If not, they were categorized as having a low level of fruit and vegetable intake (LFVI).

Linguistic validation was performed to ensure that the participant understood the questions as intended and could provide accurate answers. A focus group was asked to discuss and respond to the survey as one group. This process was repeated with the same people to test the reliability. Afterwards, a new group was interviewed to ensure the clarity of the original meaning and to develop modified questions. Accordingly, the final version of the questionnaire was produced.

### 2.5. Statistical Analysis

Prevalence was categorized by quarter and weighted to equal the adult population in Saudi Arabia, according to the General Authority of Statistics’ census report [14]. The quantitative variables are presented herein as mean and SD if they have a normal distribution, or median and range, as appropriate. The categorical variables are presented as percentages and confidence intervals (CIs), and are compared using the Pearson chi-squared test. As this study employed automated electronic data collection, there were no missing values; the QPlatform involves a data integrity check to prevent users from entering invalid data (e.g., the minimum age is 18) [6]. To explore the association between the quarters of 2020 and the behavioral risk factors, the intermediate risk factors, and the chronic diseases, a logistic regression analysis adjusted for the demographical variables (age, gender, and region) was performed. The results have been reported in accordance with the strengthening the reporting of observational studies in epidemiology (STROBE) checklist for cross-sectional studies [15].

### 2.6. Ethical Considerations

This project was approved by the Sharik institutional review board (approval no. 2021-2), in accordance with Saudi Arabia’s national research ethics law and regulations. Participant consent was obtained verbally during the interview.

## 3. Results

Of the 44,782 potential participants contacted in 2020, 30,134 from the 13 administrative regions of Saudi Arabia completed the interview, with a response rate of 67.29%. Out of the total participants, 51.2% were female. The mean age was 36.5 (SD 13.5; range: 18–99). Table 1 shows the demographic distribution by quarters.

The overall weighted prevalence of behavioral risk factors gradually decreased with AFVI and ALPA, and gradually increase with the prevalence of e-cigarette use, throughout 2020. In terms of intermediate risk factors, hypertension also showed a gradual increase over the year. The prevalence of heart disease and diabetes had increased by the end of the year. Table 2 shows the weighted prevalence of all risk factors and chronic diseases in the SHISS.

A regression analysis adjusted for age, gender, and region revealed a significant decrease in fruit and vegetable intake and physical activity, and a significant increase in e-cigarette use, over the quarters of 2020. It also revealed a significant increase in hypercholesterolemia and in hypertension. Table 3 shows the crude and adjusted odds ratios (ORs) of the association between each quarter of 2020 and all risk factors and chronic diseases in the SHISS.

## 4. Discussion

This study used data from the SHISS to explore the effects of responses to COVID-19 on some behavioral and intermediate risk factors and major chronic diseases. The behavioral risk factors displayed significant changes compared to Q1 of 2020, during which there were not yet any restrictions. Notable among these were the reductions in fruit and vegetable intake and physical activity, and the significant increase in e-cigarette use, with a slight decrease in cigarette smoking and waterpipe smoking. In terms of the intermediate risk factors observed in the SHISS, significant increases in hypercholesterolemia and hypertension were observed. By the end of the year, heart disease and diabetes displayed significant increases compared to the first quarter.

The SHISS uses a combination of leading health indicators (input indicators, which are predictive, dynamic, and fast-changing) and lagging health indicators (output indicators, which are slow-changing and based on phenomena that have already occurred). Behavioral risk factors are leading indicators, and intermediate risk factors and chronic disease are lagging indicators. We observed significant changes across the quarters, compared to Q1, in terms of behavioral risk factors—decreases in fruit and vegetable intake and physical activity, and significant increases in e-cigarette use, as well as a slight decrease in cigarette smoking and waterpipe smoking. The reduction in physical activity was due to complete or partial lockdown, the closure of sports and fitness centers, and the increase in working from home, which all directly affect the attainment of the recommended physical activity levels. Fear of contracting COVID-19 may also have affected people’s willingness to go to fitness centers or attend group fitness activities, even after the end of the lockdown [4]. A recent systematic review exploring changes in physical activity and sedentary behaviors from before to during the pandemic stated that the majority of studies reported decreases in physical activity and increases in sedentary behaviors across several populations during their respective lockdowns, including in children and patients with a variety of medical conditions [16].

The significant continuous decrease in fruit and vegetable intake is more difficult to explain. However, a study conducted in one city on the pandemic’s impact on eating habits in Saudi Arabia found that the quality and the quantity of food had been compromised [17]. Another study found that low-quality diets were linked to food insecurity, and it also reported an increased intake of savory snacks, sweets, and candies among food-secure participants [18]. This may also be related to boredom during the lockdown and increasing levels of stress and depression [4].

In terms of tobacco consumption, the slight reductions in waterpipe smoking during Q2 and Q3 are mainly due to the lockdown and the closure of waterpipe lounges. However, the slight reductions in cigarette smoking might be related to public health campaigns related to COVID-19 prevention, which highlighted smoking cigarettes as a major risk factor for contracting COVID-19 and for death by COVID-19. It seems that this slight reduction in waterpipe and cigarette smoking has shifted people towards e-cigarette smoking, which showed a significant gradual increase over the year.

Intermediate risk factors also showed some increase, especially hypercholesterolemia and hypertension in the last two quarters of the year. This increase might be related to changes in behavioral risk factors, mainly the reductions in fruit and vegetable intake and also in physical activity, as found in this study, as well as the increases in the consumption of low-quality food, as found in other studies [17,18].

Moving to the slowest lagging health indicators in the SHISS (chronic diseases), the only significant changes were observed in heart disease and diabetes, and mainly in the last quarter. Although these changes might be related to changes in behavioral and intermediate risk factors, they may also simply be another effect of the response to COVID-19 and the public’s avoidance of healthcare facilities during Q2 and Q3, which consequently delayed the diagnosis. That said, changes in chronic disease status may require longer monitoring due to the slow nature of their development.

Although this study provided some evidence of increasing behavioral and intermediate risk factors, it has some limitations. The surveillance system used is a work in progress; however, it generates frequent and unprecedented national-level data, and is currently the only one of its kind in Saudi Arabia. The short length of the questionnaire, and the community engagement model adopted by Sharik, played important roles in improving the response rate. However, we could be criticized for using quota sampling, which is associated with risks of selection bias, rather than random probability sampling. However, the costs of probabilistic sampling are significantly greater, and given this project’s aim and the types of variables it used, the risk of some generally low-level [19,20] bias was considered acceptable. In addition, using a proportionally large sample and 52 quota reduces the selection bias [19,20]. Currently, in Saudi Arabia, the only way to generate a random national-level sample is via a household survey, which also has some limitations, and was not possible under COVID-19 restrictions. However, the recruitment and sampling methods of the SHISS have been used successfully in many other national projects in Saudi Arabia [21,22,23]. In addition, the SHISS is limited in the number of variables it can include, which were selected as general public health indicators. The multi-wave survey design used in the SHISS does not allow for the monitoring of the same individuals in each wave, and can only reflect the general proportions of the indicators within the population. However, the multi-waves design, as well as the systematic, high-quality data collection process and the large sample size, make it an appropriate methodology with which to assess the effects of large-scale events on the health of the general population.

## 5. Conclusions

This study gave some evidence of the impact of responses to COVID-19 on the health of the population in Saudi Arabia. Significant reductions in fruit and vegetable intake and physical activity, and significant increases in e-cigarette use, hypertension, and hypercholesterolemia, may increase the burden of chronic disease in Saudi Arabia in the near future. Thus, the continuous monitoring of the population’s health risk factors, and early intervention, are recommended in order to prevent future increases in chronic diseases.

## Figures and Tables

**Figure 1 ijerph-18-05291-f001:**
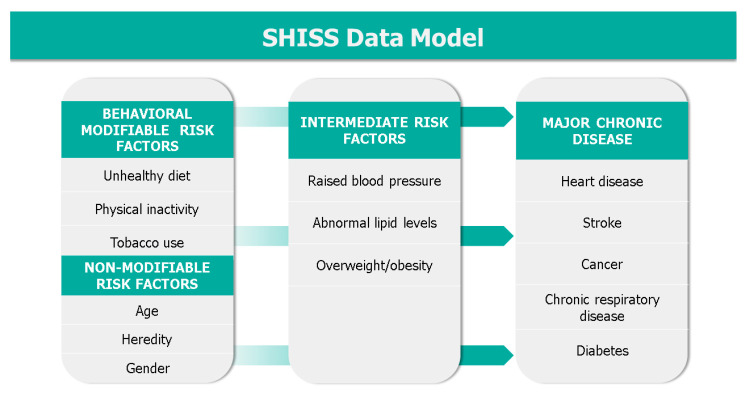
The Sharik Health Indicators Surveillance System (SHISS) data model.

**Table 1 ijerph-18-05291-t001:** Participant demographics in the sample by the quarters of 2020.

	Quarter 1 2020 (Q1) *n* (%)	Quarter 2 2020 (Q2) *n* (%)	Quarter 3 2020 (Q3) *n* (%)	Quarter 4 2020 (Q4) *n* (%)	Total *n* (%)
**Sex**					
Male	3272 (46.4)	5543 (49.1)	2583 (49.8)	3305 (50.0)	14,703 (48.8)
Female	3778 (53.6)	5746 (50.9)	2600 (50.2)	3307 (50.0)	15,431 (51.2)
**Age Groups (Years)**					
18–19	397 (5.6)	481 (4.3)	246 (4.7)	327 (4.9)	1451 (4.8)
20–29	2337 (33.1)	3862 (34.2)	1737 (33.5)	2230 (33.7)	10,166 (33.7)
30–39	1824 (25.9)	2516 (22.3)	1031 (19.9)	1211 (18.3)	6582 (21.8)
40–49	1439 (20.4)	2380 (21.1)	1139 (22.0)	1372 (20.8)	6330 (21.0)
50–59	732 (10.4)	1373 (12.2)	661 (12.8)	909 (13.7)	3675 (12.2)
60+	321 (4.6)	677 (6.0)	369 (7.1)	563 (8.5)	1930 (6.4)
**Regions**					
Asir	877 (12.4)	1028 (9.1)	382 (7.4)	542 (8.2)	2829 (9.4)
Baha	972 (13.8)	846 (7.5)	400 (7.7)	540 (8.2)	2758 (9.2)
Eastern region	930 (13.2)	870 (7.7)	401 (7.7)	542 (8.2)	2743 (9.1)
Hail	669 (9.5)	843 (7.5)	400 (7.7)	540 (8.2)	2452 (8.1)
Jazan	791 (11.2)	895 (7.9)	404 (7.8)	540 (8.2)	2630 (8.7)
Al Jouf	495 (7.0)	850 (7.5)	400 (7.7)	540 (8.2)	2285 (7.6)
Madinah	347 (4.9)	971 (8.6)	400 (7.7)	540 (8.2)	2258 (7.5)
Makkah	249 (3.5)	858 (7.6)	401 (7.7)	541 (8.2)	2049 (6.8)
Najran	25 (0.4)	808 (7.2)	400 (7.7)	122 (1.8)	1355 (4.5)
Northern border	35 (0.5)	804 (7.1)	400 (7.7)	539 (8.2)	1778 (5.9)
Qassim	300 (4.3)	820 (7.3)	401 (7.7)	540 (8.2)	2061 (6.8)
Riyadh	920 (13.0)	891 (7.9)	404 (7.8)	544 (8.2)	2759 (9.2)
Tabuk	440 (6.2)	805 (7.1)	390 (7.5)	542 (8.2)	2177 (7.2)
**Grand Total**	**7050**	**11,289**	**5183**	**6612**	**30,134**

**Table 2 ijerph-18-05291-t002:** Prevalence of chronic diseases and risk factors in the weighted sample over the quarters of 2020.

	Q1 *n* (%)	Q2 *n* (%)	Q3 *n* (%)	Q4 *n* (%)	Total *n* (%)
**Fruit and Vegetable Intake**					
AFVI	1266 (15.7)	728 (6.4)	302 (5.9)	290 (4.2)	2586 (8.2)
LFVI	6813 (84.3)	10,635 (93.6)	4849 (94.1)	6591 (95.8)	28,888 (91.8)
**Physical Activity**					
ALPA	3310 (41.0)	3007 (26.5)	1267 (24.6)	1696 (24.6)	9280 (29.5)
LLPAd	4768 (59.0)	8356 (73.5)	3884 (75.4)	5185 (75.4)	22,193 (70.5)
**Cigarette Smoking**					
Never	6658 (82.4)	9209 (81.1)	4241 (82.3)	5514 (80.1)	25,622 (81.4)
Yes, daily	1011 (12.5)	1413 (12.4)	545 (10.6)	883 (12.8)	3852 (12.2)
Yes, occasionally	409 (5.1)	740 (6.5)	365 (7.1)	484 (7.0)	1998 (6.3)
**Waterpipe Smoking**					
Never	6864 (85.0)	9587 (84.4)	4401 (85.4)	5736 (83.4)	26,588 (84.5)
Yes, daily	495 (6.1)	481 (4.2)	230 (4.5)	483 (7.0)	1689 (5.4)
Yes, occasionally	720 (8.9)	1294 (11.4)	521 (10.1)	662 (9.6)	3197 (10.2)
**E-Cigarette Smoking**					
Never	7547 (93.4)	10,361 (91.2)	4698 (91.2)	6176 (89.8)	28,782 (91.4)
Yes, daily	234 (2.9)	374 (3.3)	176 (3.4)	328 (4.8)	1112 (3.5)
Yes, occasionally	298 (3.7)	627 (5.5)	278 (5.4)	377 (5.5)	1580 (5.0)
**Hypertension**					
Yes	977 (12.1)	1615 (14.2)	819 (15.9)	1160 (16.9)	4571 (14.5)
No	7101 (87.9)	9748 (85.8)	4333 (84.1)	5721 (83.1)	26,903 (85.5)
**Hypercholesterolemia**					
Yes	884 (10.9)	1768 (15.6)	809 (15.7)	1046 (15.2)	4507 (14.3)
No	7194 (89.1)	9595 (84.4)	4343 (84.3)	5835 (84.8)	26,967 (85.7)
**Obesity**					
Yes	2180 (27.0)	2893 (25.5)	1168 (22.7)	1658 (24.1)	7899 (25.1)
No	5898 (73.0)	8470 (74.5)	3984 (77.3)	5223 (75.9)	23,575 (74.9)
**Diabetes**					
Yes	930 (11.5)	1543 (13.6)	692 (13.4)	1081 (15.7)	4246 (13.5)
No	7149 (88.5)	9820 (86.4)	4460 (86.6)	5800 (84.3)	27,229 (86.5)
**Heart Disease**					
Yes	299 (3.7)	614 (5.4)	263 (5.1)	409 (5.9)	1585 (5.0)
No	7779 (96.3)	10749 (94.6)	4888 (94.9)	6472 (94.1)	29,888 (95.0)
**Stroke**					
Yes	129 (1.6)	225 (2.0)	108 (2.1)	141 (2.0)	603 (1.9)
No	7949 (98.4)	11138 (98.0)	5044 (97.9)	6740 (98.0)	30,871 (98.1)
**Cancer**					
Yes	114 (1.4)	225 (2.0)	110 (2.1)	137 (2.0)	586 (1.9)
No	7965 (98.6)	11,138 (98.0)	5041 (97.9)	6744 (98.0)	30888 (98.1)
**Chronic Respiratory Disease**					
Yes	699 (8.7)	1172 (10.3)	427 (8.3)	586 (8.5)	2884 (9.2)
No	7380 (91.3)	10,191 (89.7)	4724 (91.7)	6295 (91.5)	28590 (90.8)
**Genetic Diseases**					
Yes	696 (8.6)	845 (7.4)	392 (7.6)	512 (7.4)	2445 (7.8)
No	7383 (91.4)	10,518 (92.6)	4760 (92.4)	6369 (92.6)	29,030 (92.2)

**Table 3 ijerph-18-05291-t003:** Crude and adjusted odds ratios (ORs) of the association between each quarter of 2020 and all risk factors and chronic diseases in the Sharik Health Indicators Surveillance System (SHISS).

Variable	Crude OR (95% CI) (*p*-Value)	Adjusted OR (95% CI) (*p*-Value)
**Fruit and Vegetable Intake**		
Q1	Reference	Reference
Q2 *	0.368 (0.335–0.405) (<0.001)	0.388 (0.351–0.429) (<0.001)
Q3 *	0.335 (0.294–0.382) (<0.001)	0.346 (0.294–0.382) (<0.001)
Q4 *	0.237 (0.208–0.270) (<0.001)	0.239 (0.209–0.274) (<0.001)
**Physical Activity**		
Q1	Reference	Reference
Q2 *	0.518 (0.488–0.551) (<0.001)	0.529 (0.497–0.563) (<0.001)
Q3 *	0.470 (0.435–0.508) (<0.001)	0.480 (0.443–0.519) (<0.001)
Q4 *	0.471 (0.439–0.506) (<0.001)	0.483 (0.450–0.519) (<0.001)
**Cigarette Smoking**		
Q1	Reference	Reference
Q2	1.096 (1.018–1.181) (0.015)	0.996 (0.919–1.079) (0.915)
Q3 *	1.006 (0.918–1.102) (0.902)	0.894 (0.810–0.987) (0.026)
Q4	1.163 (1.071–1.263) (<0.001)	1.051 (0.961–1.149) (0.275)
**Waterpipe Smoking**		
Q1	Reference	Reference
Q2	1.047 (0.967–1.133) (0.259)	0.921 (0.848–1.001) (0.053)
Q3 *	0.964 (0.874–1.064) (0.471)	0.840 (0.758–0.930) (0.001)
Q4	1.128 (1.033–1.232) (0.007)	0.993 (0.906–1.089) (0.884)
**E-Cigarette Smoking**		
Q1	Reference	Reference
Q2 *	1.372 (1.230–1.530) (<0.001)	1.279 (1.142–1.432) (<0.001)
Q3 *	1.371 (1.203–1.562) (<0.001)	1.276 (1.116–1.460) (<0.001)
Q4 *	1.620 (1.440–1.822) (<0.001)	1.531 (1.356–1.729) (<0.001)
**Hypertension**		
Q1	Reference	Reference
Q2	1.203 (1.105–1.310) (<0.001)	1.047 (0.953–1.149) (0.340)
Q3 *	1.373 (1.242–1.518) (<0.001)	1.161(1.040–1.297) (0.008)
Q4 *	1.473 (1.344–1.615) (<0.001)	1.190 (1.075–1.318) (0.001)
**Hypercholesterolemia**		
Q1	Reference	Reference
Q2 *	1.499 (1.375–1.634) (<0.001)	1.408 (1.281–1.547) (<0.001)
Q3 *	1.515 (1.368–1.679) (<0.001)	1.363 (1.219–1.525) (<0.001)
Q4 *	1.459 (1.325–1.605) (<0.001)	1.225 (1.102–1.361) (<0.001)
**Obesity**		
Q1	Reference	Reference
Q2 *	0.924 (0.866–0.986) (0.016)	0.921 (0.860–0.985) (0.016)
Q3 *	0.793 (0.731–0.860) (<0.001)	0.774 (0.711–0.843) (<0.001)
Q4 *	0.859 (0.797–0.924) (<0.001)	0.823 (0.762–0.888) (<0.001)
**Diabetes**		
Q1	Reference	Reference
Q2	1.209 (1.108–1.318) (<0.001)	1.035 (0.940–1.138) (0.485)
Q3	1.192 (1.073–1.325) (0.001)	0.966 (0.860–1.084) (0.553)
Q4 *	1.433 (1.305–1.575) (<0.001)	1.138 (1.025–1.263) (0.015)
**Heart Disease**		
Q1	Reference	Reference
Q2 *	1.486 (1.290–1.711) (<0.001)	1.257 (1.084–1.457) (0.002)
Q3	1.399 (1.181–1.657) (<0.001)	1.123 (0.941–1.340) (0.199)
Q4 *	1.645 (1.412–1.916) (<0.001)	1.279 (1.089–1.501) (0.003)
**Stroke**		
Q1	Reference	Reference
Q2	1.239 (.996–1.542) (0.054)	1.003 (0.799–1.259) (0.980)
Q3	1.313 (1.015–1.700) (0.038)	1.012 (0.775–1.323) (0.927)
Q4	1.285 (1.010–1.635) (0.041)	0.952 (0.740–1.223) (0.698)
**Cancer**		
Q1	Reference	Reference
Q2 *	1.413 (1.126–1.774) (0.003)	1.331 (1.054–1.680) (0.016)
Q3 *	1.524 (1.170–1.985) (0.002)	1.407 (1.074–1.843) (0.013)
Q4	1.422 (1.107–1.826) (0.006)	1.286 (0.995–1.661) (0.055)
**Chronic Respiratory Disease**	(<0.001)	
Q1	Reference	Reference
Q2 *	1.214 (1.100–1.339) (<0.001)	1.201 (1.086–1.328) (<0.001)
Q3	0.954 (0.841–1.082) (0.466)	0.942 (0.829–1.070) (0.356)
Q4	0.982 (0.876–1.102) (0.761)	0.964 (0.858–1.083) (0.538)

* Significant *p*-value in the adjusted column.

## Data Availability

Available from Sharik Association for Health Research upon request.

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
