# Peer review of "Exploring the Impact of COVID-19 Response on Population Health in Saudi Arabia: Results from the “Sharik” Health Indicators Surveillance System during 2020"

_ijerph, 2021, doi:10.3390/ijerph18105291_

Round 1

Reviewer 1 Report

  1.  This article is a secondary analysis based on the data of the ‘Sharik’  Health Indicators Surveillance System, but the system and the research based on the system’s data are not mentioned in the introduction, and it needs to be further improved.
  2. The English quality of your manuscript needs to be improved. Pay attention to English grammar, spelling and sentence structure. It is recommended to edit it carefully by personnel with professional English editing experience, so that readers can understand the content of the article more clearly.

Author Response

Reviewer #1:

This article is a secondary analysis based on the data of the ‘Sharik’  Health Indicators Surveillance System, but the system and the research based on the system’s data are not mentioned in the introduction, and it needs to be further improved.

Authors’ Response: Agree, more information about SHISS is now included in the introduction section.

The English quality of your manuscript needs to be improved. Pay attention to English grammar, spelling and sentence structure. It is recommended to edit it carefully by personnel with professional English editing experience, so that readers can understand the content of the article more clearly.

Authors’ Response: Thank you we used MDPI English editing service to proofread the final manuscript after this revision.

Thank you for your valuable feedback and comments

Reviewer 2 Report

The authors presented here an interesting study on  Sharik Health Indicators Surveillance System (SHISS) implemented in Saudi Arabia. It gives a good study impact of COVID-19 on the health of the population. The discussion could be improved.

Author Response

Reviewer #2:

The authors presented here an interesting study on Sharik Health Indicators Surveillance System (SHISS) implemented in Saudi Arabia. It gives a good study impact of COVID-19 on the health of the population. The discussion could be improved.

Authors’ Response: Agree, the discussion section has been improved further.

Thank you for your valuable feedback and comments.

Reviewer 3 Report

Dear Authors, 

Thank you for the opportunity to review your manuscript titled, "Exploring the Impact of COVID-19 Response on Population Health in Saudi Arabia: Results from the ‘Sharik’ Health Indicators Surveillance System during 2020." I offer the following comments for your consideration:

  1. Please provide additional information about the SHISS. For example when, how, and why was it developed? Have any significant population health findings, prior to COVID-19, occurred as a result of this phone survey? How has it been used to set health policy in Saudi Arabia, if at all. Please explain.
  2. Does this survey use both phone and land lines? If so, how might this have helped with your excellent response rate? Similarly, how can you account for the response rate being comparable in all of the quarters in 2020 and not just the later quarters of 2020 which involved the COVID-19 lockdown?
  3. The study is descriptive in nature and you clearly explain the limitations. However, it is not clear how the major chronic diseases were identified. Were these respondents already self-identified as having one of the major chronic diseases listed? If so, the findings really are about changes amongst pre-existing chronic conditions and the discussion would need to reflect this important point. If there is a mix of pre-existing and newly diagnosed respondents with chronic disease, how were the newly diagnosed identified during the lockdown? Can the pre-existing and newly diagnosed patients be distinguished in the analysis to add further meaning to the results and discussion? This point actually gets at the role of the pandemic on these indicators. For example, did one's limitations in physical activity and healthy eating practices further exacerbate pre-existing chronic illness or did they possibly contribute to an increase in new chronic disease cases? This is a critical distinction for this work that needs to be clarified and communicated for the reader. It is an interesting and informative trend to clarify.

Author Response

Reviewer #3:

Thank you for the opportunity to review your manuscript titled, "Exploring the Impact of COVID-19 Response on Population Health in Saudi Arabia: Results from the ‘Sharik’ Health Indicators Surveillance System during 2020." I offer the following comments for your consideration:

Please provide additional information about the SHISS. For example when, how, and why was it developed? Have any significant population health findings, prior to COVID-19, occurred as a result of this phone survey? How has it been used to set health policy in Saudi Arabia, if at all. Please explain.

Authors’ Response: Agree, we included more information about the SHISS in the introduction section.

Does this survey use both phone and land lines? If so, how might this have helped with your excellent response rate? Similarly, how can you account for the response rate being comparable in all of the quarters in 2020 and not just the later quarters of 2020 which involved the COVID-19 lockdown?

Authors’ Response: The survey only uses mobile phone numbers generated form the research participants database. The short nature of the questionnaire and the community engagement model adopted by Sharik play important role in improving response rate. We updated the method and discussion sections to address this point.

The study is descriptive in nature and you clearly explain the limitations. However, it is not clear how the major chronic diseases were identified. Were these respondents already self-identified as having one of the major chronic diseases listed? If so, the findings really are about changes amongst pre-existing chronic conditions and the discussion would need to reflect this important point. If there is a mix of pre-existing and newly diagnosed respondents with chronic disease, how were the newly diagnosed identified during the lockdown? Can the pre-existing and newly diagnosed patients be distinguished in the analysis to add further meaning to the results and discussion? This point actually gets at the role of the pandemic on these indicators. For example, did one's limitations in physical activity and healthy eating practices further exacerbate pre-existing chronic illness or did they possibly contribute to an increase in new chronic disease cases? This is a critical distinction for this work that needs to be clarified and communicated for the reader. It is an interesting and informative trend to clarify.

Authors’ Response: We agree with the reviewer of the importance of knowing the differences between the pre-existing conditions and new conditions, however, the SHISS is a multi-waves survey that does not monitor the same participants over time. Thus, it is not possible to generate the results suggested. However, we updated the limitation in the discussion section to clarify this point. “The multi-wave survey design used in the SHISS does not allow monitoring of the same individuals each wave and only allow for monitoring of the general proportions of the indicators within the population.”

Thank you for your valuable feedback and comments.

Round 2

Reviewer 3 Report

Thank you for addressing my comments.